# Reliability of Repeated Nordic Hamstring Strength in Rugby Players Using a Load Cell Device

**DOI:** 10.3390/s22249756

**Published:** 2022-12-13

**Authors:** Christian Chavarro-Nieto, Martyn Beaven, Nicholas Gill, Kim Hébert-Losier

**Affiliations:** 1Division of Health, Engineering, Computing and Science, Te Huataki Waiora School of Health, University of Waikato, Adams Centre for High Performance, Tauranga 3116, New Zealand; 2New Zealand Rugby, Wellington 6011, New Zealand

**Keywords:** football, muscle testing, stability, strain injuries, test-retest

## Abstract

Hamstring strain injuries are one of the most common injuries in Rugby Union players, representing up to 15% of all sustained injuries. The Nordic eccentric hamstring test assesses the maximal hamstring eccentric strength and imbalances between limbs. Asymmetries and deficits in hamstring strength between legs are commonly assessed and used as screening methods to prevent injuries which can only be proven effective if hamstring strength measures are reliable over time. We conducted a repeated-measures reliability study with 25 male Rugby Union players. Nordic eccentric strength and bilateral strength balance was assessed. Three testing sessions were undertaken over three consecutive weeks. Intrasession and intersession reliabilities were assessed using typical errors (TE), coefficient of variations (CV), and intraclass correlation coefficients (ICC). Our results showed good intrasession reliability (ICC = 0.79–0.90, TE = 26.8 N to 28.9 N, CV = 5.5% to 6.7%), whilst intersession reliability was fair for mean and the max (ICC = 0.52–0.64, TE = 44.1 N to 55.9 N, CV from 7.4% to 12.5%). Regarding the bilateral strength balance ratios, our results showed good intrasession reliability (ICC = 0.62–0.89, TE = 0.5, CV = 4.4% to 7.2%), whilst the intersession reliability for mean and max values was fair (ICC = 0.52–0.54) with a good absolute intersession reliability CV ranging from 8.2% to 9.6%. Assessing the Nordic eccentric hamstring strength and the bilateral strength balance in Rugby players using a load cell device is a feasible method to test, and demonstrated good intrasession and fair intersession reliability. Nordic eccentric strength assessment is a more practical and functional test than isokinetic; we provide data from Rugby Union players to inform clinicians, and to establish normative values in this cohort.

## 1. Introduction

Hamstring strain injuries are one of the most common injuries in Rugby Union players, representing up to 15% of all sustained injuries [1]. In England Rugby Football Union, hamstring strain is the most common occurring injury during training with a 15% incidence [2] and is the second most common injury during match play after thigh hematomas [3]. The 2019 Rugby World Cup injury surveillance data revealed lower limb injuries accounted for almost 50% of all players’ absence days. Hamstring strains were the second most common match injury after concussion in the tournament, with hamstring injuries representing 9.8% of all match injuries and causing 467 missed days [4]. As with any injury, intrinsic and extrinsic risk factors for hamstring strains have been identified [1]. Hamstring strain injuries have the highest recurrence rate of any muscle injury [5], for instance in Rugby Union, a previous hamstring injury increased the risk of a subsequent hamstring injury four-fold [6], due to residual neuromuscular inhibition, strength deficits, altered muscle tendon morphology, and modified contractile mechanics.

In Rugby Union, hamstring injuries often occur during the eccentric phase of running or kicking, and less frequently because of a direct tackle collision in the ruck position [7]. Another mechanism linked to hamstring strain injury is hamstring strength deficits and bilateral strength imbalances [8]. These imbalances in strength between muscle groups and extremities are assessed and used as screening methods in sports [8].

The Nordic eccentric hamstring test assesses the maximal hamstring eccentric strength and imbalances between limbs [9]. Nordic eccentric strength assessment is a more feasible and functional test than isokinetic (the gold standard method to measure hamstring strength). In the Australian Football Rules, an eccentric hamstring strength threshold value of 256 Newtons was established (N), below which there was a significant increase in injury risk [10]. However, this threshold could differ with different body mass (heavier and/or taller players can reach 256 N with greater ease than smaller players). In football, Buchheit et al. [11] examined the effect of body mass on hamstrings eccentric strength exercises on Nordbord. These authors estimated an increase in 4 N of eccentric hamstring strength per increase in 1 kg of body mass and provided a predictive equation of eccentric strength according to body mass (eccentric strength (N) = 4 X * Body Mass (kg) + 26.1). Values over or below 40 N (12%) of this expected value based on body mass were considered to reflect a significant imbalance or weakness in football players. Establishing such a predictive equation for Rugby Union would be of high practical value. In Rugby Union, players with a raw average eccentric strength in both limbs of less than 267.9 N did not show more risk of having a hamstring injury when compared to the stronger players; forwards were stronger than backs [7].

In the Nordic eccentric strength exercise, injured players have been reported to show an imbalance between limbs, with a mean of 17.37% which was significantly higher than the players with no injuries who displayed an imbalance mean of 10.0%. In the eccentric strength test, imbalances between legs for injured players displayed greater values compared to uninjured players. Imbalances of more than 15% between legs increased the risk of hamstring injuries by 2.4 times and imbalances of more than 20% between legs increased the risk by 3.4 times [6]. When assessing such injury risk thresholds, it is important to be cognizant of the reliability of the testing methodology to inform the minimal detectable difference and smallest worthwhile changes. Previous studies have examined the reliability of a novel device designed to measure hamstring eccentric strength and bilateral strength balance in different cohorts including Rugby Union players with a Nordic hamstring eccentric exercise in a single session and demonstrated high to moderate reliability (intra-class correlation coefficient = 0.83–0.90; typical error, 21.7–27.5 N; CV 5.8–8.5%) [9].

There is evidence to support the use of Nordic eccentric strength measures to inform practice, with strength and imbalances as useful indicators for predicting injuries [10]. We have followed the accepted methods for developing valid and reliable studies with load cell devices in Rugby Union players [12]. Although, intersession reliability for the Nordic eccentric hamstring strength with a load cell device in Rugby Union has not been examined. Given that testing with a load cell device to assess bilateral hamstring strength and imbalances, it may be a feasible surrogate method to test to the isokinetic test, we aimed to examine the intrasession and intersession reliability of Nordic eccentric hamstring strength measures in semi-professional Rugby Union players using a load cell device.

## 2. Materials and Methods

### 2.1. Study Design

A repeated-measures reliability study was conducted in semi-professional male Rugby Union players. Based on the methods described to establish minimum sample size requirements for reliability studies [13], a minimum of around 20 participants was needed when setting the acceptable reliability level at ρ_0_ = 0.40 (i.e., fair reliability threshold) and desired reliability level at ρ_1_ > 0.75 (i.e., good reliability threshold) with an α = 0.05 and β = 0.20 knowing that players were assessed on three occasions.

### 2.2. Participants

Twenty-five semi-professional male Rugby Union players (mean ± standard deviation (SD), age 23.8 ± 3.2 years, height 184.5 ± 7.2 cm and body mass 99.3 ± 9.8 kg) agreed to participate in this study. The inclusion criteria required all the participants to be free of knee and hamstring injuries in the last month that compromise maximal isometric contraction performance of the knee flexor musculature. All participants were informed of the purpose, benefits, and risks of the study through written and oral description and gave their written consent to participate prior to engaging in any activity. The study protocol was approved by the University of Waikato Human Research Ethics Committee (HREC(Health) 2019#74) and adhered to the latest Declaration of Helsinki.

### 2.3. Instrumentation

Tests were conducted using a customised device that contained two load cells (MT501 Meltron Millennium Mechatronics Limited, Auckland, New Zealand) that measured force from the right and left leg separately with a capacity of 250 kg for each load cell (error < 0.02%, sensitivity 0.08 kg). Load cells were connected via Bluetooth to a tablet (Samsung Galaxy TAB A 10. 2018 Tablet 2 GB Ram 32 GB Storage Wi-Fi Android 9.0—Black) and data were recorded at 520 Hz. The reliability of Nordic exercises with a load cell device showed good intra-session reliability (ICC = 0.79 to 0.90) and a fair reliability in the mean and max intersession (ICC = 0.52 to 0.64) Figure 1.

### 2.4. Procedures

This study assessed hamstring strength with semi-professional Rugby Union players with a load cell device. Three testing sessions were undertaken over three weeks, with each weekly session separated by seven days. The participants completed each testing session whilst performing their routine training program in a high-performance centre where they were accustoming to training. Participants knelt on a platform with the ankles attached to a load cell, and were instructed to lean forward as slowly as possible whist resisting the movement with the hamstring muscles. The device measures the eccentric force exerted by the hamstring muscle complexes whilst the muscles are lengthening under load, see Figure 2.

The same examiner supervised all tests. Before the experimental procedure, all participants completed a warm-up protocol of three submaximal repetitions of Nordic eccentric exercises with a verbal command “free fall”. For the experimental procedure participants completed three maximal effort repetitions of Nordic eccentric exercises with a verbal command “fall as far and as slow as you can”, after each repetition a 30 s rest was given between efforts. The peak force in Newtons (N) was recorded during the maximal eccentric hold.

### 2.5. Statistical Analysis 

Data are described using means ± SD. The normal distribution of variables was assessed with Shapiro-Wilks’s and d’Agostino-Pearson tests. Data were log-transformed for reliability analysis to reduce bias arising from non-uniformity of error when appropriate. The three repetitions completed during the first session were used to examine the intra-session reliability. The inter-session mean analysis was comprised of mean strength values for each trial, and inter-session maximal force analysis was comprised of the peak strength value collected during each trial. The intersession reliability reflects the stability of measures as it defines the day-to-day variability in measures, which typically needs more than one-day between measures in sport measures [14]. The reliability of intra-session and inter-session measurements was assessed using intra-class correlation coefficient (ICC), coefficient of variation (CV), typical error (TE), and mean change (Δ), and were calculated with their SD or 95% confidence limits (lower, upper) using a customized statistical Excel spreadsheets [15] in Microsoft Excel for Office MSO (Version 2111, Build 16.0.14701.20254). Relative reliability was interpreted as poor, fair, good, and excellent when corresponding ICCs were <0.40, 0.40 to 0.75, >0.75 to 0.90, and >0.90 [16]. Absolute reliability was considered good and acceptable when corresponding CVs were ≤10% and ≤20% [17,18].

Trials and repetitions were assessed for systematic error (i.e., learning effects) using a one-way repeated measures analysis of variance (RM ANOVA) using STATA (Statics/data analysis version 16.1, StataCorp, College Station, TX, USA). The Duncan method was applied in a post-hoc testing. The statistical significance level was set at *p* ≤ 0.05 for all analysis. If the assumption of sphericity was violated, the adjusted *p*-values were reported.

## 3. Results

Descriptive and reliability statistics related to intrasession isometric neck strength values and ratios are shown in Table 1. Those related to intersession mean values are displayed in Table 2, and intersession maximal values are reported in Table 3.

### 3.1. Left Leg

Nordic eccentric hamstring strength demonstrated good intra-session reliability for mean eccentric Nordics (ICC = 0.90, TE = 26.8 N and CV = 6.3%), fair inter-session reliability for mean values (ICC = 0.58, TE = 54.2 and CV = 12.5%), and good inter-session reliability for maximal values (ICC = 0.87, TE = 24.9, CV = 5.7%). There was no systematic bias across reliability analyses based on the RM ANOVAs (*p* ≥ 0.053).

### 3.2. Right Leg

Nordic eccentric hamstring strength demonstrated good intra-session reliability for mean eccentric Nordics (ICC = 0.76, TE = 28.9 N and CV = 6.7%), fair inter-session reliability for mean values (ICC = 0.64, TE = 44.1 and CV = 7.4%), and good inter-session reliability for maximal values (ICC = 0.62, TE = 44.1, CV = 9.7%). There was no systematic bias for intra-session analyses based on the RM ANOVAs (*p* ≥ 0.058). However, bias was detected for mean inter-session reliability analysis (*p* = 0.04). The post-hoc Duncan test analysis revealed a significant difference effect for trial 3 vs. 1 (*p* = 0.031) and trial 3 vs. 2 (*p* = 0.003). Bias was also detected for max inter-session reliability analysis (*p* = 0.02), and a post-hoc Duncan test analysis revealed a significant difference effect for trial 3 vs. 1 (*p* = 0.001) and trial 3 vs. 2 (*p* = 0.014).

### 3.3. Left-Right Ratio

Intra-session left-to-right ratio values demonstrated good reliability (ICC = 0.90, TE = 0.5, and CV = 5.5%), fair inter-session for mean left-to-right ratio (ICC = 0.52, TE = 0.71, and CV = 8.2%), and fair inter-session reliability for maximum force (ICC = 0.53, TE = 0.78, and CV = 9.6%). There was no systematic bias across reliability analyses based on the RM ANOVAs (*p* ≥ 0.649); however, intersession mean bias was detected (*p* = 0.02). The post-hoc Duncan test analysis revealed a significant difference effect for trial 2 vs. 1 (*p* = 0.042) and trial 3 vs. 1 (*p* = 0.012).

## 4. Discussion

We evaluated the reliability of a customized load cell device on hamstring strength and bilateral strength balance with Nordic eccentric exercises in Rugby Union players. Our results showed good intrasession reliability (ICC = 0.79–0.90), however, a fair intersession reliability—which here reflects the stability in measures—in the mean and the maximum values (ICC = 0.52–0.64). Similar to our results, the intrasession reliability of a novel load cell device using the Nordic eccentric exercises in players from different sports including Rugby Union players have shown a good test-retest intraclass reliability (ICC = 0.85–0.89) and a fair reliability for a single leg (ICC = 0.56–0.73) [9]. The authors recommended the assessment of Nordic eccentric exercises in a bilateral method to test strength and bilateral strength balance [9]. Assessing the intersession reliability of a novel Nordic hamstring eccentric strength device and compared with an isokinetic strength device in collegiate students, the test-retest for the novel device showed good to excellent intersession reliability (ICC = 0.76–0.96) for the left leg; and (ICC = 0.78–0.96) for the right leg [19]. The lower reliability in extension was thought due to variations in technique and body positioning between sessions [20]. Assessing Nordic eccentric hamstring strength involves paying attention to foot and body positions, and ensuring the movement is controlled while performing the Nordic exercise; the Nordics are subject to variations in position as well as in falling speed between sessions, which could explain the superior intrasession than intersession reliability outcomes.

Regarding values of absolute reliability, we exhibited intrasession TE values ranging from 26.8 N to 28.9 N, with a good absolute reliability CV ranging from 5.5% to 6.7%. Mean and maximal force inter-session TE values ranged from 44.1 N to 55.9 N with acceptable absolute reliability CVs ranging from 7.4% to 12.5%. Comparable to our results, the reliability of a novel load cell device using the Nordic eccentric exercises showed TE values ranged from 21.7 N to 27.5 N with CV that ranged from 5.8% to 8.5% [9]. The test-retest reliability showed of the novel device to assess hamstring eccentric exhibited good to excellent reliability between two trials with a TE value of 14.65 N for the left leg, and with a TE value of 17.29 N for the right leg [19]. Using a load cell device on a platform to test Nordic eccentric position in Rugby Union players is also subject to small but potentially meaningful variations when tested in the same session, and since the reliability of the measures were tested a week apart, they can be considered acceptable.

In a systematic literature review and meta-analysis, results showed that previous hamstring injury was a significant risk factor for hamstring injuries; additionally, previous injuries as anterior cruciate ligament, calf, and knee injuries were also linked with hamstring injuries [21]. The review highlighted the significant relationship between leg imbalances in eccentric strength and a previous hamstring strain injury, and concluded that players with a previous injury had an increased risk of sustaining another injury if they returned to play with pronounced strength imbalances between legs. In Rugby Union, isokinetic testing was the most common method of testing hamstring strength [22] and is consider the ‘gold standard’ method to test hamstring strength and bilateral strength balance [23]. Isokinetic testing is also considered the standard method to assess quadriceps strength, and hamstring to quadriceps ratio [20,24,25,26,27,28,29]. Concentric hamstring strength was examined across all isokinetic studies, with some examining eccentrics [20,24,26,27,28,29]. The majority of studies have evaluated the hamstring to quadriceps ratio (H:Q) [20,24,25,26,27,28,29,30,31,32] and others included dynamic control ratio (DCR) [20,24,26,27,28,29]. In the literature, thresholds from athletics (track-and-field) have recommended H:Q values surpass 0.6 and DCR of 1.0 [33]. When the isokinetic test was included as a tool to return to play in football, there was no significance in this strength measure as a condition to return to play after a hamstring injury [34]. The Nordic eccentric strength assessment with a portable load cell device or a Nordbord is a more feasible and functional test than isokinetic testing. However, a systematic review and meta-analysis of different devices measuring hamstring eccentric strength in different sports concluded that the Nordbord was the most common device used to test hamstring function, and advised caution when assessing hamstring peak strength and imbalances to estimate hamstring injury risk, thus, not to use it as the only tool; however, the review did recommend the Nordbord as a tool to assess in-season neuromuscular status of players [35]. Whatever the tool used, Rugby players should be monitored in their return to play progression and assessed periodically for imbalances, especially if they have sustained a previous hamstring injury.

Despite this, a study with soccer players that assessed the correlation of isokinetic dynamometry and a Nordic eccentric device, displayed poor correlations between the isokinetic test and the Nordic eccentric test (r = 0.35), with no correlation with the bilateral strength and imbalances (r = 0.037) [36]. Another study compared the Nordic hamstring eccentric strength measured with a load cell device to a Biodex isokinetic dynamometer with healthy student participants which showed a good correlation (r = 0.823–0.840). The test-retest showed good to excellent reliability of the hamstring eccentric device and concluded that the device was valid and reliable when compare with the ‘gold standard’ method [19]. However, when comparing Isokinetic dynamometry and a Nordic eccentric hamstring load cell device in healthy student athletes assessed with eccentric peak torque, bilateral strength balance and hamstring electromyography; there was a poor correlation between the two methods (r = 0.58), with lower values in the isokinetic test (∼28%), high TE (∼19%), and proportional and systematic differences. The study concluded that these devices are not appropriate to reliably determine bilateral eccentric balance [37]. When the reliability of an isokinetic Cybex Norm was assessed using hamstring strength and bilateral strength balance, a study with healthy participants found poor test-retest relative reliability of imbalance ratios (ICC = 0.69) and suggested caution when the results are interpreted in this cohort. In addition the authors recommended that in order to extrapolate these results to other populations, it was necessary to assess isokinetic testing alongside other measures to increase the reliability of bilateral strength balance ratios [38]. Furthermore, the eccentric hamstring strength measured with a Nordbord was able to identify clinically relevant bilateral strength imbalances that were not identified by isokinetic concentric testing during the first year in patients treated with an ACL reconstruction using a hamstring tendon autograft [35]. It could be valuable to compare isokinetic test outcomes to Nordic eccentric strength outcomes specifically in Rugby Union players to determine their interchangeability, which could confirm the validity of using a Nordic load cell device for testing in Rugby Union players.

Regarding the bilateral strength balance ratios, our results showed good intra-session reliability (ICC = 0.62–0.89) with good absolute intra-session reliability CV ranging from 4.4% to 7.2%. The inter-session reliability for mean and maximal values was fair (ICC = 0.5–0.54) with good absolute inter-session reliability CV ranging from 8.2% to 9.6%. We found hamstring strength values ranging from 398 to 506 N with bilateral strength balance ratios of 0.98 to 1.0. A study with a load cell device with Nordic eccentric exercises in semi-professional Rugby Union players, showed a peak value of 387.9 ± 81.5 in both legs, and a bilateral strength balance difference of 10 ± 9.8% [6]. Our results demonstrated greater measures of Nordic eccentric hamstring strength values compared to measures in different sport athletes tested with a cell load device ranging from 321 to 391 N, and similar bilateral strength balance ratio ranging from 0.92 to 0.97 [9]. Regarding strength and imbalances assessed with Nordic eccentric load cell devices, the studies by Wiesinger [36] and Impellizzeri [37] agreed that measures of hamstring eccentric strength and bilateral strength balance were acceptable to detect large strength changes. Importantly, it was suggested that these changes are particularly important to clinicians implementing rehabilitation programs, but not appropriate to detect small changes induced by training strategies in athletes or healthy individuals. The identification of normative hamstring strength and bilateral strength balance values is of the utmost importance for clinicians interested in in Rugby Union to screen and to determine the relationships between specific hamstring strength, imbalances, and hamstring injury risk.

## 5. Conclusions

Assessing the Nordic eccentric hamstring strength and the bilateral strength balance in Rugby players using a load cell device is a feasible method to test and demonstrated good intra-session and fair intersession relative reliability. The absolute reliability is good intra-session and acceptable inter-session. Here, we provide data from Rugby Union players to inform clinicians, and to establish normative values in this cohort. Additional research with Nordic eccentric load cell devices to improve intersession reliability and stability of measures ensuring an initial familiarisation session, testing players off-season and with further attention to foot and body positions, as well as ensuring the movement is controlled while performing the Nordic exercise is advised.

## Figures and Tables

**Figure 1 sensors-22-09756-f001:**
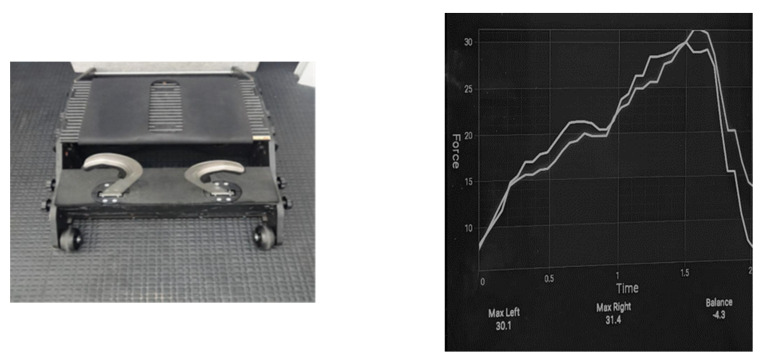
Illustration of the load cell device and the real-time visual display of peak hamstring strength (N) and bilateral strength balance (%) values.

**Figure 2 sensors-22-09756-f002:**
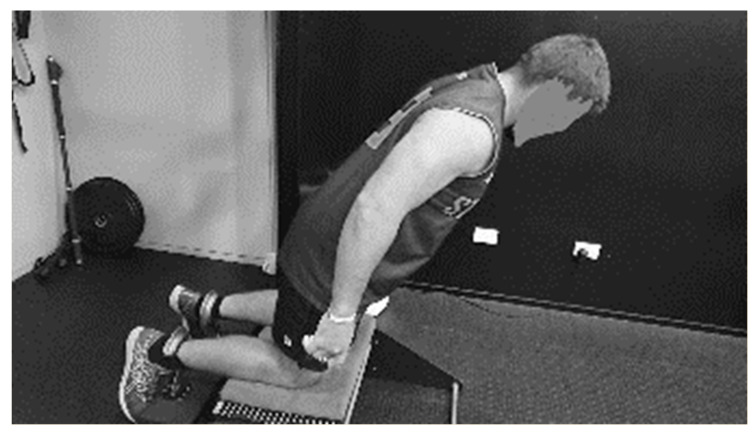
Illustration of Nordic eccentric test performed using a load cell device.

**Table 1 sensors-22-09756-t001:** Descriptive and reliability statistics related to intrasession isometric neck strength values. Values include mean, standard deviation, and 95% confidence intervals (upper, lower) for left leg, right leg, and balance examined. *p*-value from repeated measures analysis of variance. CI: confidence interval, CV: coefficient of variation, ICC: intraclass correlation coefficient, SD: standard deviation, T1: trial 1, T2: trial 2, T3, trial 3, TE: typical error.

	Mean Nordic Strength (SD)			Δ Eccentric Strength (SD)			Reliability Statistics			
	Trial 1	Trial 2	Trial 3	Trial 1–2	Trial 2–3	Trial 1–3	ICC [95% CI]	TE (N) [95% CI]	CV (%) [95% CI]	*p*-Value
**Left leg flexion** **(Newton)** **Right leg flexion** **(Newton)** **Bilateral ratio**	472.9 (67.9) 453.7 (48.7) 1.04 (0.13)	466.8 (61.4) 445.9 (65.1) 1.05 (0.1)	466.4 (75.4) 453.7 (63.2) 1.0 (0.13)	−6.1 (40.3) −7.6 (33.8) −0.02	3.7 (42.9) 10.3 (48.3) 0.02	6.5 (29.2) −1.2 (39.6) 0.01	0.90 [0.82–0.95] 0.76 [0.6–0.87] 0.79 [0.62–0.89]	26.8 [22.7–33.2] 28.9 [24.5–35] 0.5 [0.4–0.6]	6.3 [5.2–8.4] 6.7 [5.5–8.9] 5.5 [4.4–7.2]	0.783 0.454 0.221

**Table 2 sensors-22-09756-t002:** Descriptive and reliability statistics related to intersession of Nordic eccentric hamstring strength values (mean of three trials). Values include mean, standard deviation, and 95% confidence intervals (upper, lower) for left leg, right leg, and balance examined. *p*-value from repeated measures analysis of variance. CI: confidence interval, CV: coefficient of variation, ICC: intraclass correlation coefficient, SD: standard deviation, T1: trial 1, T2: trial 2, T3, trial 3, TE: typical error.

	Mean Nordic Strength (SD)			Δ Eccentric Strength (SD)			Reliability Statistics			
Variable	Trial 1	Trial 2	Trial 3	Trial 1–2	Trial 2–3	Trial 1–3	ICC [95% CI]	TE (N) [95% CI]	CV (%) [95% CI]	*p*-Value
**Left leg flexion** **(Newton)** **Right leg flexion** **(Newton**) **Bilateral ratio**	470.2 (65.9) 449.6 (54.7) 1.04 (0.11)	454.1 (114.7) 449.7 (83.4) 1.0 (0.13)	478.3 (74.8) 492.1 (54.7) 1.0 (0.13)	−13.9 (83.5) 5.6 (65.1) 0.05	38 (91.2) 50.7 (70.8) 0.02	−19.0 (44.2) −44.9 (48.2) −0.06	0.59 [0.40–0.75] 0.64 [0.55–0.72] 0.52 [0.32–0.70]	54.2 [46.6–66.4] 44.1 [38–53.9] 0.71 [0.6–0.8]	12.5 [10.7–15.5] 7.4 [5.8–10.5] 8.2 [7.0–10.1]	0.175 0.003 0.023

**Table 3 sensors-22-09756-t003:** Descriptive and reliability statistics related to intersession of Nordic eccentric hamstring strength values (maximal value from three trials). Values include mean, standard deviation, and 95% confidence intervals (upper, lower) for left leg, right leg, and balance. *p*-value from repeated measures analysis of variance. CI: confidence interval, CV: coefficient of variation, ICC: intraclass correlation coefficient, SD: standard deviation, T1: trial 1, T2: trial 2, T3, trial 3, TE: typical error.

	Mean Nordic Strength (SD)			Δ Eccentric Strength (SD)			Reliability Statistics			
Variable	Trial 1	Trial 2	Trial 3	Trial 1–2	Trial 2–3	Trial 1–3	ICC [95% CI]	TE (N) [95% CI]	CV (%) [95% CI]	*p*-Value
**Left leg flexion** **(Newton)** **Right leg flexion** **(Newton)** **Bilateral ratio**	491.1 (67.7) 474.3 (57.9) 1.03 (0.11)	420.2 (190.5) 472.3 (87.2) 1.0 (0.14)	398.8 (215.3) 506.6 (65.6) 0.98 (0.08)	−10.9 (81.9) 0.5 (6.8) 0.04	35.7 (95.6) 4.6 (7.4) 0.01	−20.5 (51.7) −0.4 (4.4) −0.05	0.56 [0.37–0.73] 0.62 [0.44−0.77] 0.54 [0.33–0.71]	55.9 [48–68.7] 44.1 [37.253.9] 0.78 [0.65–1.0]	12.5 [10.6–15.4] 9.7 [8.2–11.9] 9.6 [7.9–12.3]	0.252 0.021 0.121

## Data Availability

Not applicable.

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
