# Peer review of "Reliability of Repeated Nordic Hamstring Strength in Rugby Players Using a Load Cell Device"

_sensors, 2022, doi:10.3390/s22249756_

Round 1

Reviewer 1 Report

Comments and Suggestions for Authors

The paper entitled « Reliability of repeated Nordic hamstring strength in rugby players using a load cell device » presents experimental works aimed at assessing the intra- and intersession reliability of Nordic eccentric hamstring strength assessment compared to that of isokinetic assessments, in the frame of hamstring strain injuries.

The background of the study is extensively described, as well as the statistical analysis performed on the experimental results. Results show that using a load cell device is not only more convenient that classical isokinetic measurements, but also that data may be very useful in designing rehabilitation programs.

The background, materials, methods and results are clearly exposed.

Comments:

Extensive editing is required: several sentences should be rephrased for clarity and a number of typing mistakes are present in the manuscript. Figure and Table numbers are also wrong and should be corrected.

Reviewer 2 Report

Tables are not clear.

Discussion needs improvement (there are small number of references).

Conclusion is the summary.

References should be supplemented (there are only 24 positions).

Reviewer 3 Report

The manuscript does not have sufficient depth. Almost half of the document consists of tables and visuals. Literature research is insufficient. It is difficult to understand what the research question and hypotheses are. It is not possible to come across an approach that improves or facilitates the currently used methods.

Reviewer 4 Report

TITLE: Reliability of repeated Nordic hamstring strength in rugby players using a load cell device.

Date: 21.11.2022

1.- Does the title clearly reflect its content?

Yes, it's very clear. With special attention to the instrument used

2.- Is the organization correct and the presentation clear?

Yes, the structure is very clear and makes reference to other instruments to measure the research problem

3.- Are the results and conclusions justified?

They are well presented. With multiple references and very concrete and clear examples.

4.- Are the bibliographical references adequate or are fundamental works on the subject matter of this article missing?

Adequate References, 20-25. Okay

5.- Recommend its publication:

Without any modification.....................X

With variations.....................................

Restructured and modified.....................

Must be rejected.....................................

6.- Suggested modifications or comments that you wish to make.

Very honest when they refer to the lack of studies to continue obtaining data.

The decision of the editors is key in this case, I insist, the work is very good.

Round 2

Reviewer 3 Report

The manuscript's revised form look good enough.